# It Takes Two to Tango! Protein–Protein Interactions behind cAMP-Mediated CFTR Regulation

**DOI:** 10.3390/ijms241310538

**Published:** 2023-06-23

**Authors:** Alessandra Murabito, Janki Bhatt, Alessandra Ghigo

**Affiliations:** 1Department of Molecular Biotechnology and Health Sciences, Molecular Biotechnology Center “Guido Tarone”, University of Torino, 10126 Torino, Italy; janki.bhatt@unito.it; 2Kither Biotech S.r.l., 10126 Torino, Italy

**Keywords:** CFTR, CFTR interactors, cAMP signaling

## Abstract

Over the last fifteen years, with the approval of the first molecular treatments, a breakthrough era has begun for patients with cystic fibrosis (CF), the rare genetic disease caused by mutations in the gene encoding the cystic fibrosis transmembrane conductance regulator (CFTR). These molecules, known as CFTR modulators, have led to unprecedented improvements in the lung function and quality of life of most CF patients. However, the efficacy of these drugs is still suboptimal, and the clinical response is highly variable even among individuals bearing the same mutation. Furthermore, not all patients carrying rare CFTR mutations are eligible for CFTR modulator therapies, indicating the need for alternative and/or add-on therapeutic approaches. Because the second messenger 3′,5′-cyclic adenosine monophosphate (cAMP) represents the primary trigger for CFTR activation and a major regulator of different steps of the life cycle of the channel, there is growing interest in devising ways to fine-tune the cAMP signaling pathway for therapeutic purposes. This review article summarizes current knowledge regarding the role of cAMP signalosomes, i.e., multiprotein complexes bringing together key enzymes of the cAMP pathway, in the regulation of CFTR function, and discusses how modulating this signaling cascade could be leveraged for therapeutic intervention in CF.

## 1. Introduction

Cystic fibrosis (CF) is the most common life-threatening inherited disorder among the Caucasian population, affecting more than 100,000 people worldwide. It is caused by mutations in the gene encoding the cystic fibrosis transmembrane conductance regulator (CFTR), a plasma membrane chloride (Cl^−^) and bicarbonate (HCO_3_^−^) channel expressed in epithelial cells, whose dysfunction leads to impaired mucus hydration and clearance [1]. As a consequence, CF impacts the functionality of multiple organs, although the disease primarily affects the respiratory system and digestive tract. If, on the one hand, gastrointestinal symptoms can be limited by pancreatic enzyme replacement therapy, on the other hand, treating lung dysfunction remains the most challenging aspect of disease management. The respiratory pathology of CF is characterized by persistent airway obstruction that progresses from early onset mucus plugging in small airways to chronic airway inflammation and recurrent respiratory infections, ultimately leading to lung destruction and respiratory failure, which is the primary cause of death among CF patients [2].

Despite the cloning of the *CFTR* gene in 1989, significant progress in the development of CFTR-targeting therapies has only occurred in the past decade, when CFTR modulators, the first molecular treatments directly targeting the underlying molecular defect of CF, were approved. In particular, a game-changing event for CF patients has been the FDA approval in 2019 of Elexacaftor-Tezacaftor-Ivacaftor (ETI), a triple-combination therapy able to substantially improve lung function, especially in patients carrying at least one F508del allele, who represent 80% and 90% of the total European and U.S. CF population, respectively [3,4,5]. However, ETI still has some limitations. Indeed, patients with rare genotypes, who account for about 20% of the European CF population, are currently ineligible for treatment with these modulators, since not all CFTR mutant proteins are responsive to these compounds [6]. Furthermore, ETI can only modulate up to 50–60% of wild-type channel mutant F508del-CFTR activity [7,8], indicating that alternative and/or add-on therapeutic approaches are still urgently needed for CF. 

One possibility to unveil new druggable targets is to pinpoint CFTR interactors and the related signaling pathways involved in regulating channel processing, trafficking, stability, and function, all mechanisms that can be dysregulated in CF patients [9]. A promising candidate is the signaling cascade orchestrated by 3′,5′-cyclic adenosine monophosphate (cAMP), since this second messenger can regulate multiple aspects of CFTR function, from channel open probability to its localization in the plasma membrane [10]. More importantly, as discussed briefly in the following paragraph, cAMP is compartmentalized in space and time thanks to the formation of localized signaling complexes, also known as cAMP signalosomes, anchored to specific subcellular compartments or organelles within the cell, and thus represents an ideal target for fine-tuned therapeutic approaches [11].

## 2. cAMP Signaling Compartmentalization

cAMP works alongside other second messengers to relay extracellular signals, primarily through hormones and neurotransmitters, to the intracellular environment, ensuring that a given external signal is converted into the appropriate cellular response. The molecular constituents as well as the different events behind the activation of the cAMP pathway are well-characterized. The binding of the ligand to a Gs-protein-coupled receptor (GPCR) located at the plasma membrane triggers the activation of heterotrimeric G proteins which, in turn, stimulate adenylate cyclase (AC) to produce cAMP from ATP [12]. cAMP in turn activates protein kinase A (PKA), the most widely researched serine/threonine kinase composed of two catalytic (Cα, Cβ, and Cγ) and two regulatory (RI and RII) subunits [13]. At supraphysiological levels, two cAMP molecules bind each of the PKA-R subunits to induce the dissociation of the tetramer, leading to the release of the PKA-C subunits which are then free to phosphorylate a multitude of intracellular substrates before being rapidly recaptured [14]. Instead, upon physiological cAMP stimulation, the catalytically active PKA holoenzyme remains intact, indicating that the range of action of the kinase is restrained [15]. Alternatively, cAMP can activate other effectors, including cyclic nucleotide-gated ion channels (CNGs) [16], and the exchange protein directly activated by cAMP (EPAC) [17]. The signal is then switched off by GPCR desensitization, de-phosphorylation of effectors mediated by phosphatases, or by the activity of phosphodiesterases (PDEs), the only enzymes capable of hydrolyzing cAMP into its respective 5′-monophosphate. In addition, the intracellular levels of the second messenger can be reduced by ABC transporters, namely multidrug resistance-associated proteins (MRPs), which export cAMP from the cell [18]. 

Since the identification of cAMP as the molecule responsible for the ancestral fight-or-flight response to catecholamines at the end of the 1950s [19], several other cellular functions have been found to be regulated by cAMP generation. One example of distinct cellular responses triggered by the same second messenger in the same cell came from the early finding in the late 1970s that, in the heart, isoproterenol, but not prostaglandin, can increase cardiac contractility, even if both hormones induce the synthesis of a similar level of intracellular cAMP [20]. Subsequent studies led to the discovery that different receptors can trigger the production of different pools of cAMP that, in turn, generate distinct responses within the same cell thanks to their confinement in spatially restricted subcellular compartments, spanning from the nano- to the micro-meter scale in size [21,22,23]. These distinct cAMP pools are generated through the formation of localized multiprotein complexes that allow restricted production and destruction of the second messenger in specific subcellular regions as well as selective involvement of distinct signal transducers, thus preventing the activation of undesired effectors [24]. One of the major determinants of cAMP compartmentalization is the hydrolysis of the second messenger by PDEs that, with more than 100 isoforms displaying different kinetics and cellular localizations, spatially regulate the duration and amplitude of the signals, controlling cAMP diffusion to neighboring compartments in which the activation of PKA and other unwanted cAMP effectors is prevented [25]. The activity of cAMP signalosomes, and in particular of PKA, is also spatially regulated by A-kinase anchoring proteins (AKAPs). AKAPs are a diversified group of scaffold proteins that, by binding directly to the regulatory subunits of PKA, recruit the holoenzyme to specific cellular structures, such as cytoskeletal components, close to their phosphorylation targets [11]. In addition to PKA, AKAPs can interact with other players in cAMP signaling and, even with different signal transduction machineries, act as crossroads for different signaling pathways [11]. The CFTR channel itself is part of a cAMP signalosome that includes the GPCR β2-adrenergic receptor (β2AR), ACs, and PDEs. In the following paragraphs, we will describe in detail cAMP-related multiprotein complexes controlling CFTR activity and localization.

## 3. cAMP-Related Macromolecular Complexes Regulating CFTR Opening

cAMP is recognized as the master regulator of CFTR activity. cAMP elevation in the subcortical compartment leads to the activation of PKA and PKA-mediated phosphorylation of the CFTR is required to increase the channel open probability and to allow efflux of Cl^–^ anions [10]. In epithelial cells, the cAMP pool responsible for CFTR function is primarily produced upon the activation of a specific GPCR, namely the β2AR channel (Figure 1). This protein co-localizes with the CFTR at the apical membrane of polarized lung epithelial cells (Calu-3), and it has also been found to co-immunoprecipitate with the channel itself [26,27]. The physical protein-protein interaction (PPI) between the anion channel and the β2AR is coupled with CFTR activation since βAR agonists, such as isoproterenol (β1AR and β2AR agonist) and albuterol (β2AR-selective agonist), can induce a concentration-dependent increase in short-circuit current (I_SC_), which is sensitive to CFTR-selective inhibition in polarized epithelial cells [26]. It is also notable that the β2AR-dependent CFTR opening can be stimulated only upon the interaction between the two proteins, since the removal of the PDZ-binding motif of CFTR, which abolishes the physical interaction between the channel and the receptor, specifically reduces Cl^−^ efflux after β2AR stimulation in vitro [26]. The relevance of β2ARs to CFTR activation is corroborated by the observation that β2AR agonists induce the swelling of intestinal organoids, a gold-standard model in CF research, indicating that these molecules can potently activate the wild-type (wt) CFTR [28]. Again, in intestinal epithelial cells, it has been shown that AC6 plays a central role in regulating CFTR function upon GPCR activation (Figure 1). AC6, the most abundant AC isoform in the gut, can be detected as a CFTR interactor in the intestinal mucosa of mice and in colon epithelial cells [29]. In further support of the key role of AC6 in producing the cAMP pool responsible for CFTR activity in the gut, Thomas and colleagues showed that the AC activator forskolin failed to induce the swelling of intestinal organoids derived from AC6 knockout mice [29]. Another well-characterized interactor with the CFTR protein is PKA, one of the main effectors of the second messenger cAMP. PKA phosphorylates the CFTR protein on several serines mainly located within the regulatory R domain of the channel (Figure 1), an unstructured polypeptide sequence with a predominant inhibitory function. In its unphosphorylated state, the R domain intercalates between the two nucleotide-binding domains (NBDs), preventing their dimerization and CFTR opening [30], whereas phosphorylation of residues S422, S660, S795, and S813, can release the channel from its closed conformation, increasing its open probability up to 100-fold [31]. Unlike the aforementioned serines, whose post-translational modification is considered as being activating, some residues such as S737 and S768 have been proposed to be either activating or inhibitory, depending on contextual modifications of other phosphorylation sites [32]. In this regard, the phosphorylation of different serine residues can be interdependent, as seen in S795 and S813. The latter can be phosphorylated only when S795 has already been post-translationally modified, showcasing S813 phosphorylation as a limiting step in CFTR activation [33]. 

A recent study suggested that, in addition to the phosphorylation of key serine residues, the simple binding of PKA to the CFTR contributes to the activation of the channel [31]. Mihályi and colleagues demonstrated that, while the post-translational modification of the unstructured R domain is required for sustained channel activation, the binding of the kinase to the CFTR is by itself able to release the unphosphorylated R domain from its inhibitory position, thus leading to Cl^−^ secretion [31]. PKA was found to interact with the CFTR channel and in particular to co-immunoprecipitate with both the catalytic and type II regulatory subunits of the kinase in the early 2000s [34]. The early finding that this interaction is abrogated upon treatment with Ht31, a peptide disrupting the interaction between AKAPs and the RI or RII subunits of PKA, indicated that a PKA-anchoring protein is responsible for the association between the kinase and the CFTR [34]. Subsequent studies identified ezrin as the scaffold protein bringing PKA into close proximity with the Cl^−^ channel (Figure 1). Ezrin can co-immunoprecipitate with both the RII subunit of the kinase and the CFTR, and the inhibition of its AKAP function with the Ht31 peptide reduces the cAMP-activated membrane conductance of Calu-3 cells by 83%, indicating that the association between ezrin and PKA is key to the kinase-dependent regulation of CFTR gating [34]. 

The binding of ezrin to the channel is not direct, rather it involves an adaptor known as the Na+/H+ exchanger regulatory factor 1 (NHERF1), which can interact with the C terminus of the CFTR via its PDZ domains, serving as a bridge between ezrin and the Cl^−^ channel (Figure 1) [35]. In particular, in immortalized bronchial epithelial cell (16HBE14o-) monolayers, the deletion of NHERF1 PDZ domains, as well as of its FERM-binding domain responsible for the association with ezrin, abrogates CFTR-dependent Cl^−^ efflux, pointing to a crucial role of NHERF1 in the formation of the CFTR multi-protein complex which is key to the regulation of the activity of the channel [36]. In support of this data, the absence of NHERF1 completely abolishes duodenal bicarbonate secretion in response to clenbuterol-mediated β2AR stimulation in vivo [37]. Another level of NHERF1-dependent CFTR regulation resides in the fact that this adaptor protein competes with the PDZ domain-containing protein Shank2 for CFTR binding [38]. Different from NHERF1, Shank2 is a negative regulator of cAMP-mediated channel function (Figure 1). Patch clamp experiments in the whole cell configuration revealed that the overexpression of Shank2 in CFTR-expressing mouse embryonic fibroblasts (NIH 3T3) decreased CFTR activity by 50% upon forskolin-mediated AC activation, as a consequence of a significant reduction in the cAMP-induced phosphorylation of the channel [39]. This negative effect of Shank2 overexpression on CFTR activity is instead lost upon treatment with rolipram, a PDE4-specific inhibitor, suggesting a functional association between the cAMP-hydrolyzing enzyme and Shank2, which was confirmed by an increase in Shank2/PDE4 physical association upon forskolin treatment [38]. 

As further validation of the presence of PDE4 in the cAMP signalosome of the CFTR, this cAMP-hydrolyzing enzyme was found to interact with the channel in airway epithelial cells. The use of chamber measurements revealed that PDE4 inhibition increases CFTR activity after βAR stimulation and significantly reduces the time required for the channel to return to baseline activity levels [40]. In addition, the expression of a catalytically inactive PDE4, which acts by displacing the endogenous protein from the cAMP signalosome, significantly augments PKA-mediated phosphorylation of the CFTR, pointing out the key role of PDE4 in controlling CFTR via regulation of local cAMP/PKA levels [40]. Another key regulator of this CFTR-anchored pool of PDE4 is the AKAP phosphoinositide 3-kinase γ (PI3Kγ). By anchoring PKA to PDE4, PI3Kγ favors its PKA-mediated phosphorylation and consequent activation (Figure 1) [41]. Displacement of the PI3Kγ-anchored pool of PKA inhibits the activity of subcortical PDE4, leading to a localized cAMP elevation responsible for PKA-mediated CFTR phosphorylation and gating, which unveils PI3Kγ as the regulator of a cAMP microdomain central to epithelial fluid secretion in the airways [41]. Another PDE isotype, namely PDE8, was found to regulate CFTR function. Selective pharmacological PDE8 inhibition in primary bronchial epithelial cells stimulates CFTR-dependent ion transport, both under basal conditions and after pre-treatment with cAMP-elevating agents, suggesting that this negative regulator of the channel may also be a component of the cAMP signalosome regulating the CFTR [42]. In addition to PDEs, MRP channels are also responsible for limiting the pool of cAMP that regulates CFTR function, with MRP4 being the isoform found to functionally and physically associate with the CFTR protein [43,44]. In particular, MRP4 inhibition leads to a restricted increase in cAMP levels in the subcortical compartment where the transporter interacts with the CFTR channel via the PDZ Domain Containing 1 protein (PDZK1, also known as CAP70), suggesting that MRP4 modulates the pool of cAMP responsible for CFTR opening (Figure 1) [44]. This model is strengthened by the observation that MRP4 inhibition increases CFTR-mediated I*_SC_* in response to an endogenous stimulus triggering a compartmentalized cAMP increase, while it does not affect the current elicited by the global cAMP elevation induced by forskolin [44]. Finally, disruption of MRP4/PDZK1 protein-protein interaction significantly attenuates CFTR-mediated currents activated by MRP4 inhibition in gut epithelial cells, confirming the role of MRP4 as one of the crucial CFTR interactors regulating the function of the channel [44].

## 4. cAMP-Related Macromolecular Interactions Influencing CFTR Localization at the Plasma Membrane

Several studies highlighted the pivotal role of cAMP-related CFTR interactors in regulating the localization of the channel at the plasma membrane, which is the result of a tight balance between its secretion, endocytosis, and recycling [45]. The highly conserved 14-3-3 proteins, which are able to bind and stabilize proteins in their Ser/Thr-phosphorylated state, augment CFTR trafficking to the plasma membrane by binding to its R domain upon PKA-mediated post-translational modifications [46,47]. Although there are numerous 14-3-3 protein interactions where multi-phosphorylated proteins interact with 14-3-3, such as p53 and BAD [48,49], the CFTR:14-3-3 interaction is unusual, owing to the presence of nine phosphorylated binding sites in the CFTR R domain, the largest known number of binding sites for any 14-3-3 protein interaction thus far [47]. In addition to these nine motifs of CFTR that, upon phosphorylation, can bind 14-3-3 proteins, 14-3-3 proteins themselves are known to naturally form dimers consisting of two adjacent amphipathic grooves, capable of simultaneously binding multiple proteins [47,50,51]. Consequently, the numerous binding possibilities make the nature of CFTR:14-3-3 interaction complicated to decode. In the recent past, several studies have attempted to explicate the CFTR:14-3-3 protein-protein interaction. For example, Liang and colleagues found that among the seven isoforms of 14-3-3 that exist, 14-3-3β is the one with a reasonable affinity for most of the PKA phosphorylation sites in the R domain of CFTR, with pS768, pS795, and pS813 being the key binding sites behind 14-3-3:CFTR interactions [46]. Additionally, Stevers and colleagues showed that pS768 occupies one of the two amphipathic grooves of the 14-3-3 dimer, while a secondary, weaker binding site occupies the other amphipathic groove [47]. On this basis, the authors hypothesized that stabilizing this secondary binding could potentially enhance the 14-3-3:CFTR interaction. They also showed that treatment with fusicoccin, a naturally occurring phytotoxic terpenoid stabilizing the interaction between 14-3-3 proteins and their binding partners, increases the affinity of CFTR binding to 14-3-3 by nine-fold [47]. Further confirmation of the importance of 14-3-3 proteins in regulating CFTR trafficking comes from pulse-chase experiments in HEK293 cells showing that 14-3-3 proteins influence CFTR stability by reducing CFTR degradation. Accordingly, overexpression of 14-3-3β increases steady-state CFTR protein levels while its knockdown leads to reduced total and plasma membrane CFTR levels [46].

If, on the one hand, cAMP/PKA-mediated phosphorylation of the CFTR regulates its interaction with 14-3-3 proteins and thus channel trafficking to the plasma membrane, then on the other hand, the cAMP signalosome can influence the stability of the channel once it has reached the cell surface [52,53]. This process is mainly regulated by a cAMP effector other than PKA, namely EPAC1. While low concentrations of cAMP activate PKA, which in turn phosphorylates the R domain of the CFTR leading to channel opening, higher concentrations of cAMP can trigger EPAC1, which in turn promotes CFTR stability at the plasma membrane by attenuating its endocytosis [54,55]. Because EPAC1 lacks the PDZ domain necessary for direct association with the CFTR, an adaptor protein is required to mediate the EPAC1-CFTR interaction [56]. It is well-established that activated EPAC1 binds to the CFTR via NHERF1, which, as mentioned above, is a scaffold protein that contains two PDZ domains (PDZ1 and PDZ2) [57,58], which can bind to the CFTR C-terminal PDZ-binding motif (Figure 2). While the PDZ1 domain of NHERF1 can readily bind to this CFTR motif, the PDZ2 domain is blocked from binding to CFTR due to its conformation [52,53]. The binding of ezrin to NHERF1 triggers a conformational change in the protein that is responsible for the exposure of its PDZ2 domain and, ultimately, to the interaction with the PDZ-binding motif of CFTR. Together with NHERF1, the AKAP ezrin contributes to CFTR stability on the cell surface by cross-linking the CFTR to the actin cytoskeleton, which is fundamental for maintaining the CFTR in highly restricted compartments at the plasma membrane and delaying its endocytosis (Figure 2) [35,59]. In this respect, a recent study unveiled two dynamic regulators of the actin cytoskeleton which influence CFTR stability as a result of EPAC1 activation by cAMP, namely CAPZA2 (capping actin protein of muscle Z-line alpha subunit 2) and INF2 (inverted formin-2) (Figure 2) [60]. CAPZA2 is a highly conserved capping protein that blocks the addition or loss of actin monomers by capping the barbed ends of actin filaments [60,61]. In the context of CFTR regulation, Santos and colleagues observed that stimulation of human CF bronchial epithelial (CFBE) cells expressing wt-CFTR with a cAMP analog increases the interaction between EPAC1 and CAPZA2. As a consequence of the enhanced interaction of the two proteins, CAPZA2 can further potentiate the CFTR anchoring to the plasma membrane by stabilizing the actin cytoskeleton (Figure 2) [60]. While CAPZA2 stabilizes the barbed ends of actin filaments hence promoting CFTR stability, INF2 is a unique formin among the members of this class of proteins with the ability to accelerate actin polymerization but also depolymerization. INF2 binds laterally to the barbed ends of an actin filament promoting its severing, thus negatively influencing CFTR stability at the plasma membrane (Figure 2) [60]. Different studies have highlighted the existence of a strong competition between capping proteins and formins which is crucial for the tight regulation of actin dynamics, which corroborates the finding that CAPZA2 and INF2 have opposite effects on CFTR stability, given their different roles in regulating actin filaments [60,62]. Overall, these studies emphasize the role of cAMP in promoting CFTR stability through the activation of EPAC1 and of its downstream effectors serving as actin regulators. Future studies are awaited to elucidate the therapeutic significance of CAPZA2 overexpression or INF2 knockdown as potential novel strategies for augmenting CFTR stability at the plasma membrane [60].

## 5. Targeting cAMP Signaling for Therapeutic Purposes

CFTR modulators, such as ETI, currently represent the standard of care for many CF patients. Over the past few years, these molecules have brought a significant improvement in CF care thanks to their ability to target specific molecular defects of mutant CFTR channels, with a particular focus on those of the F508del variant [63,64]. The F508del mutation in the *CFTR* gene causes protein misfolding, preventing CFTR channel trafficking to the plasma membrane and thus leading to its premature degradation. Furthermore, the small amount of F508del-CFTR protein that escapes from the degradative process and succeeds in reaching the plasma membrane displays stability defects as well as impaired channel gating [65]. Some of the molecular defects of F508del-CFTR can be rescued by CFTR modulators, which can be either “correctors”, such as Elexacaftor (VX-661), Tezacaftor (VX-445), and Lumacaftor (VX-809), molecules able to restore protein folding and cellular trafficking, or “potentiators”, such as Ivacaftor (VX-770), that instead address the abnormal channel gating [66]. However, as already mentioned above, the high variability of patients’ response to these drugs and their suboptimal efficacy as well as patients’ still-limited eligibility for these treatments underlie the need for alternative and/or add-on therapeutic approaches for CF [6,7,8]. 

Growing evidence suggests that cAMP modulation could have additive therapeutic effects to those of clinically approved CFTR modulators. A compartmentalized increase in intracellular cAMP in the vicinity of the CFTR can, on the one hand, boost cAMP/PKA-mediated channel activation while, on the other hand, favoring cAMP-dependent channel stability [10,54]. Because cAMP levels in the CFTR compartment are the result of a tight balance between its production downstream of β2ARs, its degradation by local PDEs, and MRP4-mediated cAMP export, pharmacological modulation of these processes represent promising therapeutic avenues (Table 1). Small molecules activating β2ARs have been available for clinical practice since the 80s, thanks to their effect as bronchodilators [67]. 

For the same reason, these molecules are currently prescribed to 95% of CF patients, but their ability to improve the function of CFTR mutants was not studied until recently. Vijftigschild et al. first demonstrated that β2AR agonists are potent inducers of CFTR function in homozygous CFTR-F508del organoids and highly differentiated primary CF airway epithelial cells after rescue of CFTR trafficking by small molecules [28]. Conversely, another study evaluating the impact of chronic or excess β-agonist exposure showed a reduction in CFTR-dependent conductance following chronic (72 h) exposure to albuterol of immortalized CFBE cells (CFBE41o-) expressing the F508del-CFTR, and primary human airway epithelial cells (HAECs) from F508del donors [68]. These findings demonstrate that chronic treatment with β2AR agonists limits CFTR activation in airway epithelial cells and leads to more than 60% reduction in the activation of pharmacologically corrected F508del-CFTR [68]. This negative effect of β2AR agonists on CFTR function can be overcome if AC stimulation is bypassed by treating cells with a cell-permeable cAMP, indicating that the β2AR-induced CFTR dysfunction can be attributed to reduced production of intracellular cAMP downstream of the β2AR-AC axis [68]. These results, which highlight the potential detrimental interaction between CFTR modulators and β2AR-agonists, suggest that long-term stimulation of these receptors can have negative clinical implications potentially relevant to most CF patients and that new therapeutic regimens have to be taken into account [68].

Alternative pharmaceuticals that elevate cAMP levels in the proximity of the CFTR by acting downstream from β2AR-AC, including PDE inhibitors, have been tested [41,69]. Of the 11 PDE families found in humans, PDE4 is the one that predominantly localizes in close proximity to CFTR in airway epithelial cells and physically interacts with the channel, suggesting that selective PDE4 inhibition can be exploited to raise intracellular cAMP levels locally in order to augment cAMP-dependent CFTR activity [69]. Notably, the clinically approved PDE4 inhibitor Roflumilast synergizes with the potentiator VX-770, while the dual PDE3/4 inhibitor RPL554 was shown to stimulate rare class III and IV CFTR mutants, characterized by defective channel gating and conductance, respectively [69]. However, PDE4 inhibitors that have been developed in the last decades induce several side effects, such as nausea, emesis, diarrhea, and headache, mainly due to systemic inhibition of the PDE4 isotype. Thus, these compounds have a low therapeutic index, that importantly limits the dose that patients can receive and, consequently, the clinical efficacy [70]. 

Another possible strategy for the manipulation of cAMP signaling is the inhibition of ATP binding cassette transporter C4 (ABCC4), or MRP4, the cAMP exporter that is functionally and physically coupled with CFTR via the scaffolding protein PDZK1 [73,74]. Pharmacological inhibition of ABBC4 using a small molecule inhibitor, such as MK-571, attenuates the extracellular transport of cAMP, resulting in its intracellular accumulation in gut and airway epithelial cells [73]. In addition to regulating wt-CFTR function [44], MK-571 was found to enhance the response of human airway epithelial Calu-3 cells to VX-770 [73], strengthening the data discussed above, which show the synergistic effects of cAMP with existing CFTR modulators. However, common MRP4 inhibitors such as MK-571, dipyridamole, and indomethacin have limited use due to their considerable off-target effects linked to inhibition of MRP1, MRP2, MRP3, MRP5, thus implying the need for more specific ABCC4 inhibitors. In this regard, Cheung and colleagues screened and identified two new highly specific ABCC4 inhibitors, namely Ceefourin-1 and Ceefourin-2, which demonstrated low cellular toxicity and high stability both in vivo and in vitro [71]. Although these initial studies showed a good potential for these molecules, to our knowledge these drugs have not been developed further for cAMP modulation-based treatment in CF.

## 6. Targeting Protein-Protein Interactions Central to cAMP-Dependent CFTR Regulation for Therapeutic Purposes

In addition to the usage of molecules targeting the cAMP signaling pathway, which may carry the limitations discussed above, an intriguing alternative approach is provided by PPI modulators, namely molecules designed to specifically block or stabilize PPI implicated in CFTR regulation (Table 1). Although targeting PPIs may be challenging, primarily because of the dynamic and large protein interfaces involved, some PPI modulators have been approved for clinical use for cancer treatment and are already available on the market, while others are currently under clinical evaluation [75,76], underlining the therapeutic potential of these molecules. PPI modulators can have different mechanisms of action as they can: (i) act as competitive antagonists and thus prevent the formation of protein complexes by directly interacting with the surface binding site of the target protein; (ii) work as allosteric modulators via binding to a protein region that is not within the surface binding site of the two protein partners; (iii) trigger a conformational change in the target protein responsible for enhancing the interaction with the binding partner. The activity of fusicoccin, which increases the binding affinity of CFTR to 14-3-3 by stabilizing their interaction, represents a good example of the potential applicability of PPI modulators to CF therapy [47]. Several PPI modulators under preclinical evaluation for pathologies other than CF, spanning from cancer to neurological diseases, specifically target PDZ-dependent interactions [72]. Notably, most of the PPIs crucial for the regulation of the activity and the plasma membrane stability of CFTR rely on PDZ domains that, with their very low sequence similarity, are considered promising drug targets [72]. Accordingly, small molecules and peptides are currently being developed to disrupt the interaction between the CFTR and the negative regulator of channel half-life, the PDZ-containing protein CAL (CFTR-associated ligand). Instead, other PPI modulators could be used to trigger a localized and sustained increase in the cAMP pool responsible for CFTR opening. These include disruptors of the PDZ-mediated interaction of CFTR with its negative regulators, such as molecules interfering with the association between the channel and Shank2, or with the interaction between MRP4 and PDZK1. Finally, stabilizing the PDZ-mediated NHERF1:CFTR complex with a potential PPI stabilizer could positively affect both the function and the plasma membrane localization of the channel [72].

In addition to modulators targeting PDZ domains, another valuable approach could be the use of PPI disruptors of AKAP-dependent complexes. Several AKAP disruptors have been developed over the past 20 years which have helped elucidate the mechanism of action of these scaffold proteins and, at the same time, demonstrated significant therapeutic potential in different pathologies [11]. Among these is the PI3Kγ mimetic peptide that, by disrupting the interaction between the AKAP PI3Kγ and PKA, inhibits PI3Kγ-associated phosphodiesterases (PDE4B and PDE4D) and in turn increases subcortical cAMP levels, subsequently enhancing CFTR channel activity. Interestingly, the peptide works synergistically with the triple combination of correctors (VX-661, VX-445) and a potentiator (VX-770) as well as in combination with the corrector VX-809 [41]. 

If, on the one hand, there is a growing interest in developing PPI modulators for therapeutic purposes, primarily because of their exquisite target specificity, then on the other hand the design of these molecules may be challenging. This is the case for PPIs that are characterized by flat and large interaction interfaces as well as for PPIs involving proteins for which high-resolution structures are not available. Furthermore, peptidic PPI modulators may carry the limitation of having poor stability in biological fluids/tissues and being immunogenic, thus implying the need of important chemical optimization work [76]. For all of the reasons above, only a few PPI modulators has reached the clinical use hitherto and, especially in the field of CF, additional preclinical and clinical studies are required to definitively prove the feasibility of this type of therapy.

## 7. Conclusions

Altogether, the studies summarized in this review article highlight the importance of cAMP-related macromolecular complexes in the control of CFTR activity and plasma membrane localization as well as the possibility of manipulating cAMP signaling or cAMP signalosomes for therapeutic purposes in CF. Among cAMP-modulating agents already approved for clinical use are PDE4 inhibitors and β2AR agonists, which are used for chronic obstructive pulmonary disease (COPD) and asthma [70], respectively, and could be repurposed for CF treatment. Notably, these molecules could ensure additional therapeutic effects other than CFTR modulation, considering the anti-inflammatory properties of Roflumilast (PDE4 inhibitor) [77] and the bronchodilator effects of short- and long-acting β2AR agonists [78,79]. Nevertheless, the controversial effects of the latter on CFTR activity, as reported in preclinical studies, advocate the need of additional investigations with these molecules in CF models. 

An intriguing alternative approach for the manipulation of cAMP-dependent CFTR regulation is provided by PPI modulators, although the bench-to-bedside translation of these molecules is still a long way off. The efficacy of some PPI modulators has been studied in clinical trials in recent years, but primarily as anticancer agents [76]. In CF, the most advanced compound is a PI3Kγ mimetic peptide, whose efficacy as a therapeutic has been proven in gold-standard CF preclinical models [41], and which is currently under clinical development. Other PPI modulators potentially useful in CF have not been investigated in pathologically relevant settings or are still to be conceived based on the available proof-of-concept mechanistic studies. Despite the fact that the design of PPI modulators could pose challenges [76], these molecules could offer the opportunity to target specific cAMP signalosomes involved in the fine regulation of CFTR function. Nevertheless, because some CFTR mutants may exhibit a reduced sensitivity to cAMP stimulation compared to others [80], additional studies are needed in the future to clarify to what extent different CF mutations impact the organization and the activity of cAMP signalosomes involved in CFTR regulation, and thus the sensitivity of different CFTR mutants to therapeutic interventions based on cAMP modulation. 

## Figures and Tables

**Figure 1 ijms-24-10538-f001:**
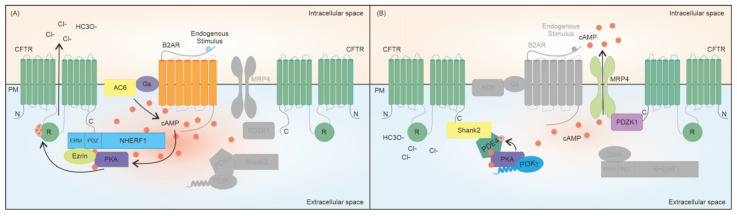
Schematic representation of CFTR interactors of the cAMP signalosomes that positively (**A**) and negatively (**B**) regulate CFTR-mediated anion secretion. (**A**) Endogenous stimulation of β2ARs leads to Gs-mediated activation of AC6, which is responsible for cAMP production. cAMP in turn activates PKA that, by phosphorylating the R domain of the CFTR channel, leads to anion secretion. PKA localizes in the vicinity of the CFTR via ezrin, an AKAP that interacts with the C terminus of the channel via the adaptor protein NHERF1. (**B**) PDE4, which interacts with the CFTR protein via Shank2, is anchored to PKA by the AKAP PI3Kγ, which favors PKA-mediated phosphorylation and consequent activation of the cAMP-hydrolyzing enzyme, switching off β2AR-mediated activation of the CFTR channel. The signal can also be terminated by the cAMP exporter MRP4, which interacts with the CFTR via the scaffolding protein PDZK1. PM: plasma membrane.

**Figure 2 ijms-24-10538-f002:**
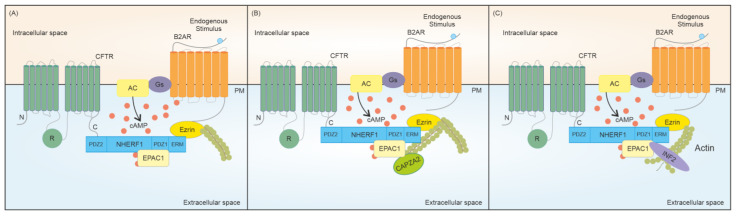
Schematic representation of cAMP-related macromolecular complexes that influence CFTR stability at the plasma membrane. (**A**) EPAC1, which is activated by the pool of cAMP produced upon stimulation of the β2AR-Gs-AC axis, binds to the CFTR channel through the scaffold protein NHERF1. This interaction anchors the anion channel to the actin cytoskeleton via the adaptor protein ezrin, stabilizing the channel at the plasma membrane. (**B**) The CFTR-NHERF1-Ezrin-EPAC1 complex is further sustained at the membrane thanks to CAPZA2, which stabilizes the barbed end of actin filaments. (**C**) Activated EPAC1 recruits INF2 that increases the number of depolymerizable pointed actin ends, thus limiting CFTR anchoring to the plasma membrane. PM: plasma membrane.

**Table 1 ijms-24-10538-t001:** Overview of the existing and proposed therapeutic strategies targeting cAMP signaling as well as cAMP-mediated complexes regulating CFTR function and stability.

No.	Therapeutic Class	Mechanism of Action	Drug Name	Status of Therapeutic Strategy	References
cAMP signaling modulators					
1	β2AR agonists	Stimulation of β2 adrenergic receptors; stimulation of cAMP production	Albuterol, Salbutamol	Acute β2AR agonists are potent inducers of CFTR function in homozygous CFTR-F508del organoids and highly differentiated primary CF airway epithelial cells after rescue of CFTR trafficking by small molecules. Chronic exposure to β2AR agonists limits CFTR activation in primary human airway epithelial cells expressing the F508del-CFTR.	[28,68]
2	PDE4 inhibitors	Selective inhibition of phosphodiesterase 4; inhibition of cAMP degradation	Roflumilast (oral), RPL554 (inhaled)	Clinical use of oral PDE4 inhibitors is hampered by low therapeutic index. Improved therapeutic index possible with inhaled PDE4 inhibitors.	[69,70]
3	MRP4 inhibitors	MRP4 inhibition; blockade of cAMP extracellular export	MK-571, dipyridamole, and indomethacin	Limited use and low efficacy due to off-target effects such as inhibition of MRP1, MRP2, MRP3, MRP5.	[43,71]
			Ceefourin-1 and Ceefourin-2	Highly specific MRP4 inhibitors with low toxicity and high efficacy, currently not being developed for CF.	[71]
**PPI modulators**					
1	PPI stabilizers	NHERF1:CFTR complex stabilization	n/a	NHERF1 overexpression has been shown to increase CFTR stability at the plasma membrane in CFBE cells.	[36]
		CFTR-14:3:3 stabilization	Fusicoccin	Fusicoccin has been shown to increase the affinity of CFTR for 14:3:3 proteins by nine-fold, thus stabilizing their interaction.	[47]
2	PPI disruptors	Disruption of Shank2 and CFTR	n/a	Shank2 overexpression has been shown to CFTR activity by 50% in NIH3T3 cells, hence disruption of Shank2/CFTR interaction could be a potential strategy for enhancing CFTR activity	[11,38,72]
		Selective inhibition of the A-kinase anchoring protein function of PI3Kγ; inhibition of cAMP degradation	PI3Kγ mimetic peptide	Shown to synergize with the CF modulator therapy ETI in primary human bronchial epithelial cells homozygous for F508del-CFTR; currently under clinical development.	[41]

## Data Availability

Not applicable.

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
