# Peer review of "It Takes Two to Tango! Protein–Protein Interactions behind cAMP-Mediated CFTR Regulation"

_ijms, 2023, doi:10.3390/ijms241310538_

Round 1
Reviewer 1 Report
The article provides a comprehensive overview of the significance of cAMP-related macromolecular complexes in controlling cystic fibrosis transmembrane conductance regulator activity and localization at the apical surface of epithelial cells. The authors highlight the importance of understanding the intricate signaling pathways involved in CFTR regulation for developing effective therapeutic strategies for cystic fibrosis.
One of the key findings discussed in the article is the potential clinical utility of molecules targeting the cAMP signaling pathway. Specifically, the authors mention the approval of PDE inhibitors for the treatment of chronic obstructive pulmonary disease and their potential repurposing for CF treatment. This highlights the translational relevance of the research in identifying existing drugs that could be beneficial for CF patients.
The article also introduces an intriguing alternative approach involving peptides and peptidomimetics that selectively block interactions between A-kinase anchoring proteins, PKA (protein kinase A), PDEs, and other PKA substrates. By fine-tuning these protein-protein interactions, the aim is to modulate CFTR regulation mediated by cAMP. The potential of this class of therapeutics is highlighted, with its exceptional efficacy and specificity compared to traditional small molecules, making it an area of growing interest in various pathological conditions. This is a point where I particularly ask the authors to detail more and add a schematic review of all therapeutic approaches.
However, the authors emphasize the need for further research to fully understand the impact of different CF mutations on the cAMP signalosomes involved in CFTR regulation. Additionally, the sensitivity of different CFTR mutants to cAMP modulation may influence the effectiveness of cAMP-dependent therapeutic approaches.
All limitations of the clinical scenarios should be discussed in order to have a clear view of the real world situation.
Overall, this review article successfully synthesizes existing knowledge and highlights the importance of studying cAMP-related macromolecular complexes in CFTR regulation. The article provides valuable insights into potential therapeutic avenues and raises important questions for future research. It is a valuable theoretical resource for researchers and clinicians working towards developing innovative treatments for cystic fibrosis that could be improved by adding the previously mentioned aspects.
I saw a few typos and there are also some phrases that should be rewritten in order to express logically their meaning. The authors should carefully read the entire text and correct it.
Reviewer 2 Report
Murabito et al
The authors present a review outlining the CFTR pathway, it's interacting partners and cAMP signaling therein. They discuss CFTR function, present therapeutic strategies, cAMP signaling, and binding partners of CFTR. They argue that targeting protein-protein interactions of this pathway is a way forward for targeting CFTR for therapy.
I think that the review is comprehensive and should be accepted. I have one major comment only: The way therapeutics shall be designed against protein-protein interactions and how exactly that will bring CFTR-based disorders under control is only scantily discussed. They do so a little bit between lines 401-410. However, major discussion is about cAMP-based regulation strategies that could be distinct from targeting protein-protein interactions of CFTR. Either the relationship between cAMP-based activation of CFTR should be more clearly linked with binding partners of CFTR, or these topics should be treated separately (like the authors have mostly done) and it should be said that therapeutics could target cAMP signaling or binding partners of CFTR. Currently, how cAMP signaling is linked to binding partners is unclear except more obvious proteins like PKA.
Minor point: Line 354, Using, not Ussing
